# A street corner education: Stop and search, trust, and gendered norms among adolescent males

Ben Bradford[1]*, Krisztián Pósch[1], Jonathan Jackson[2,3], Paul Dawson[4]

**1** Centre for Global City Policing, Department of Security and Crime Science, University College London, London, United Kingdom, **2** Department of Methodology, London School of Economics and Political Science, London, United Kingdom, **3** University of Sydney Law School, Sydney, Australia, **4** Mayor's Office for Policing and Crime, London, United Kingdom

* ben.bradford@ucl.ac.uk

**Data Availability Statement:** All relevant data are within the paper and its Supporting Information files.

**Funding:** The survey we use was funded and fielded by the London Mayor's Office for Policing

## Abstract

Police stop and search activity has consistently been shown to affect the opinions, attitudes and behaviours of those subject to it. For young people in particular this can be an important moment in which they learn about and orientate themselves towards law, authority, and the exercise of power. Drawing on work into procedural justice and legal socialisation, we build on the premise that stop and search has, in practice, more to do with the imposition of authority on the streets than the accurate targeting tool of crime-control activity. We consider the link between experiences of stop and search, trust in the police, exposure to or involvement in gangs and violence, and the extent to which male adolescents hold abusive and controlling gendered beliefs regarding sexuality and intimate partner relations. Using data from a survey of Londoners aged 14–16, we find support for the notion that adolescent males' procedurally unjust stop and search experiences are associated with lower levels of trust in the police, higher levels of involvement in and exposure to gang-related activities, and believing it is acceptable to harass females in public space and control intimate partners. We conclude with the idea that unfair stop/searches can signal that it is 'OK' to abuse power.

## Introduction

### A street corner education: Stop and search, trust, and gendered social norms

Police in the United Kingdom use stop and search (S&S) powers for the detection and prevention of crime, but also to assert order and control marginal populations [1, 2]. Among police and policymakers there is a widespread belief that S&S is an effective way of achieving these types of aims—S&S is a 'go to' power for those seeking to address violent and other crimes, and one of the more important ways in which officers seek to assert and enforce behavioural norms in the communities they police.

and Crime (MOPAC). The current paper constitutes secondary analysis of this data, and Bradford, Jackson and Posch received no specific funding for this work; nor did the universities they work for. However, Dawson is an employee of MOPAC, and was involved in the design and management of the survey as part of his normal work role.

**Competing interests:** The authors have declared that no competing interests exist.

Yet, there is little UK-based evidence that S&S has much influence on offending behaviour aside from drug use, and then only weakly [3, 4]. Even when positive effects are identified, studies regularly conclude that very large increases in S&S are likely to be required to produce small decreases in crime [5]. Notably, the 'hit rate' of S&S, which refers to whether a search results a criminal justice outcome of some kind (which may or may not be related to the object actually searched for), tends to be low. For example, Ashby [6, p.7] reports that between April and September 2020, 76% of S&S conducted in London resulted in no further action (i.e. no crime of any kind was uncovered).

By contrast, there is significant evidence that assertive, enforcement-lead police activity like S&S is associated with more negative opinions, attitudes and emotions among those who experience it [7, 8]. In particular, S&S has been linked to lower levels of police legitimacy [9], more cynical attitudes towards the law [10], and increased law-breaking behaviour [11]. Such statistical effects seem to arise primarily from stops that are experienced as procedurally unjust, i.e. unfair in terms of interpersonal treatment and decision-making [12].

In this paper we contribute to research on the effects of S&S on people's attitude and opinions. We focus on the possibility that, when judged procedurally unjust, being stop/searched could shift normative attitudes concerning gendered interpersonal interactions and relationships in a negative direction. Attitudes towards appropriate interpersonal behaviour are shaped most importantly during childhood and adolescence [13, 14], in the home, school and within peer groups [15–17]. But theories of legal socialisation suggest that experiences of legal authorities can also have an effect. Our judgements about what is 'right or wrong' are linked to our interactions with formal legal actors [18, 19]. Starting from the premise that S&S is, in practice, generally ineffective in securing compliance with the law, we explore whether experiences of this type of policing feed in other ways into the set of personal, social and institutional processes that shape how people think about right and wrong. In particular, we consider the potential role of procedurally unjust regulatory interactions with police officers in fostering counter-normative attitudes regarding gender and intimate partner relations among male adolescents.

To the best of our knowledge, this paper comprises the first empirical assessment of the link between S&S experiences and views on gendered relations and sexuality. At the heart of ongoing debates about the violence, sexism and misogyny that young women experience are the attitudes and norms of young men that normalize and trivialise harassment, abuse and controlling behaviour in public space and within intimate partner relationships [20–22]. There is existing evidence that the experience of discriminatory policing and consequent distrust in the police can play a role in knife-carrying and related gang-related activity [23, 24]. This may be due, in part, to a neutralisation of conventional values [25, p.28] caused by perceiving those who represent the law as unjust. Subjectively unfair S&S experiences—and the resulting lack of trust, and exposure to and involvement in gang activity—may thus be associated with the loosening of normative constraints on behaviour [26–28].

Drawing on research on procedural justice and legal socialisation, we argue that when adolescent males experience police stops as procedurally unjust, their sense of unfairness is linked to the experience of, and exposure to, counter-normative behaviour (like knife-carrying and hate-based messaging on social media) and also, relatedly, their beliefs about personal and intimate gender relations that run counter to notions of respect and equality between males and females. Gendered social norms define the practices that are expected of males and females, typically in binary, heteronormative ways [29]. Starting from the premise that certain stereotypical forms of hegemonic masculinities coalesce around notions of toughness, competition and self-protection, as well as sexual objectification in public space and control within intimate-relationships, we consider the idea that gendered norms (based on gendered dominance

that render it acceptable to sexually harass in public space and to control romantic partners) could be activated and/or strengthened when normative constraints are loosened by the subjective experience of procedural injustice.

Using data from a survey of young people in London, the 2018 Mayor's Office for Policing and Crime (MOPAC) Youth Survey, we focus on male adolescents to explore (a) the association between S&S and trust in police, (b) the association between S&S, trust and exposure to, or involvement in, violence-related and gang-related behaviours, and (c) the associations between experiences of policing, exposure to violence, and gendered social norms. We find evidence of links between S&S experienced as procedurally unjust, trust in the police, gang-related behaviours and exposure, and normative attitudes relating to gender relations and intimate partnerships.

At the threshold, it is important to note that we only have cross-sectional survey data at hand. Given that there has been no work (to our knowledge) on this specific topic, and given that the cross-sectional nature of the survey data means that we could test a variety of different theoretical models, our analysis should therefore be consider generative in nature. Drawing on extant research, we develop and provide an initial test of a framework that should next be addressed using longitudinal and/or experimental designs. The fact that our data fit this model should be taken merely as an invitation for future research to unpick the ordering and putative causal pathways suggested.

The rest of the paper proceeds as follows. After outlining the literature on policing and legal socialisation, we present a conceptual map that links (a) S&S as a particular power and mode of policing, to (b) trust in the police, to (c) exposure to, and experience of, gang-related activities, to (d) gendered social norms. We present six organising hypotheses. Sections on data and methods, results and discussion follow.

## Policing and (legal) socialisation

Over more than three decades, research on procedural justice has demonstrated that the fairness of a legal authority's behaviour—across dimensions of neutrality, respect, dignity, openness and voice—is central to explaining variation in its perceived legitimacy and in the law-related attitudes and behaviours of people with whom it interacts [12, 30–35]. A sense of trust and inclusion generated by procedural justice encourages and maintains a fit in people's minds between 'their' values and norms and those they ascribe to the authority. This alignment of norms and values then motivates behaviour congruent with the expectations of authorities, like cooperation with the police (e.g. reporting crimes) and complying with their orders. In the procedural justice literature [36, 37] this is most commonly stated in terms of the generation of legitimacy and legal compliance among adults. Yet, it is plausible that such alignment constitutes a process of socialisation (particularly when the people in question are children) involving the inculcation of a wider set of attitudes toward legal authorities, the law, and behaviours proscribed by law among those who experience procedural justice in their interactions with legal authorities.

*Legal socialisation* operates through two complementary processes [18, 19, 38]. First, it involves the development of orientations towards the law, encompassing normative expectations regarding formal social control and the individual's stance towards the societal role of law enforcement. Second, behavioural norms underlying the rule of law are internalised, and these then inform social conduct and promote self-regulation. On this account, as young people grow up and begin to encounter authority figures outside the family and school, particularly legal actors, they draw lessons from their direct and vicarious interactions concerning how they themselves should behave.

This has been described as a structured process of learning analogous to a formal curriculum. Justice and Meares [39] argue that the 'overt curriculum' of the justice system—in some liberal democracies at least—communicates messages of democratic accountability, neutrality and fairness, and identifies the goals of law enforcement as the promotion of public safety, safeguarding social and political order, and so on. This curriculum is communicated by formal education in school, and media and fictional accounts of law and policing, as well as the public statements and behaviours of justice agents. People's encounters with representatives of the justice system, on the other hand, embody a 'hidden curriculum' that has a co-educating, experiential, role. Here, often outside formal and public arenas (with the street stop as perhaps the paradigmatic encounter), people learn directly about the way the law works in practice and about the values actually enacted by police officers and other legal authorities.

The overt and covert curriculums may align around principles of procedural and substantive justice (that the law is applied correctly, with the aim of achieving appropriate specific ends). But when they contradict each other, and when justice actors do not live up to the normative expectations set out in the formal institutions of law and the values that justice organizations are meant to promote, people can come to feel that that they are objects of suspicion, a class of 'problem people' to be managed, disciplined and/or excluded—i.e. 'over-policed' [40] and 'under-protected' [41]. The 'education' they receive is quite different to that set out in the overt curriculum, and they draw commensurate lessons about their place in society and the hold that societal expectations should have on them.

An important part of this process is the procedural fairness of officer behaviour. Encounters with police are often characterised as 'teachable moments' because they signal the status, values, norms, and obligations of both citizens and legal authorities [12, 39, 42]. The police represent 'the law' [36]; procedurally just officer behaviour communicates that the values embedded in the law are appropriate, and that legal authorities enforce the law in a similarly appropriate manner. Crucially, the early experiences of young people may be fundamental in 'setting' their views of the police and the legal system [43], and research has found that young people are more sensitive to cues of procedural justice than adults [44].

Alongside legal socialisation, a parallel process may also be in play through which young people learn from interactions with police what constitutes appropriate behaviour for those with power over others. An important social norm across many different contexts is that those with power should behave respectfully, in a neutral fashion, allowing voice, and explaining their decisions when asked to do so [45] Yet, the behaviour of power-holders can serve to undermine this norm—perhaps even communicate the idea that other types of *counter*-normative, denigrating and aggressive behaviour are appropriate. If a police officer treats a young person with a lack of respect, in ways that serve to humiliate and exclude them, it is plausible to suggest that the young person may begin to internalise the values inscribed in these behaviours, and possibly act on them in the future—in other words, they may start to treat those around them with a similar fashion, and to abuse the power they have over, for example, smaller or weaker children. This is not a process of legal socialisation *per se*, because it does not rely on the behaviours concerned being proscribed by law. At stake here is the question of how people should behave towards each other in a general sense, particularly in situations where there is an imbalance of power.

It follows that when authority figures such as police officers behave in ways that develop trust, this can promote a sense that they—and the laws they enact and enforce—are well intentioned, aligned with a set of shared values, and effective in resolving conflicts. Development of trust and inclusion also generates a sense that police power is used appropriately, and an understanding of what appropriate use of power *is*. In turn, this may influence the individual's views on what constitutes appropriate behaviour. Conversely, when people feel police are *not*

behaving as they should (i.e. unfairly), this can undermine trust, and they might draw the lesson that the values embedded in the law are not appropriate guides for behaviour, not least because those charged with enforcing the law and maintaining a proper normative framework appear to be breaching them. Under such conditions, people's commitment to the law, and to social norms more widely, may be diminished. As we argue below, if S&S experiences (viewed as disrespectful and biased) can signal to young people that it is 'ok' to abuse interpersonal power, this may remove normative constraints, meaning that male adolescents are more likely to internalise unhealthy gendered norms regarding unequal male-female relationships.

## The current study

By way of orientation, Fig 1 provides an overarching conceptual map for our empirical work. On the left-hand side is the experience of S&S. The next two layers are trust in police and personal involvement in, and social exposure to, gang-related activities. At the right-hand side is a particular set of gendered normative beliefs regarding sexually aggressive behaviour in public space (e.g. staring at people that they find attractive and making sexist comments) and within-relationship control (e.g. checking a partner's phone to see who they have been interacting with). Putting all this together, a test of the model allows us to assess whether police stops experienced as unfair and discriminatory are linked to loosened normative constraints on behaviour, which finds expression here through evidence of the internalisation of stereotypically masculine norms regarding what is or is not appropriate behaviour in the spheres of sexuality and relationships.

**Stop and search: A moment of legal socialisation.**   S&S is one of the most high-profile powers vested in the police, and one that routinely courts controversy concerning its efficacy

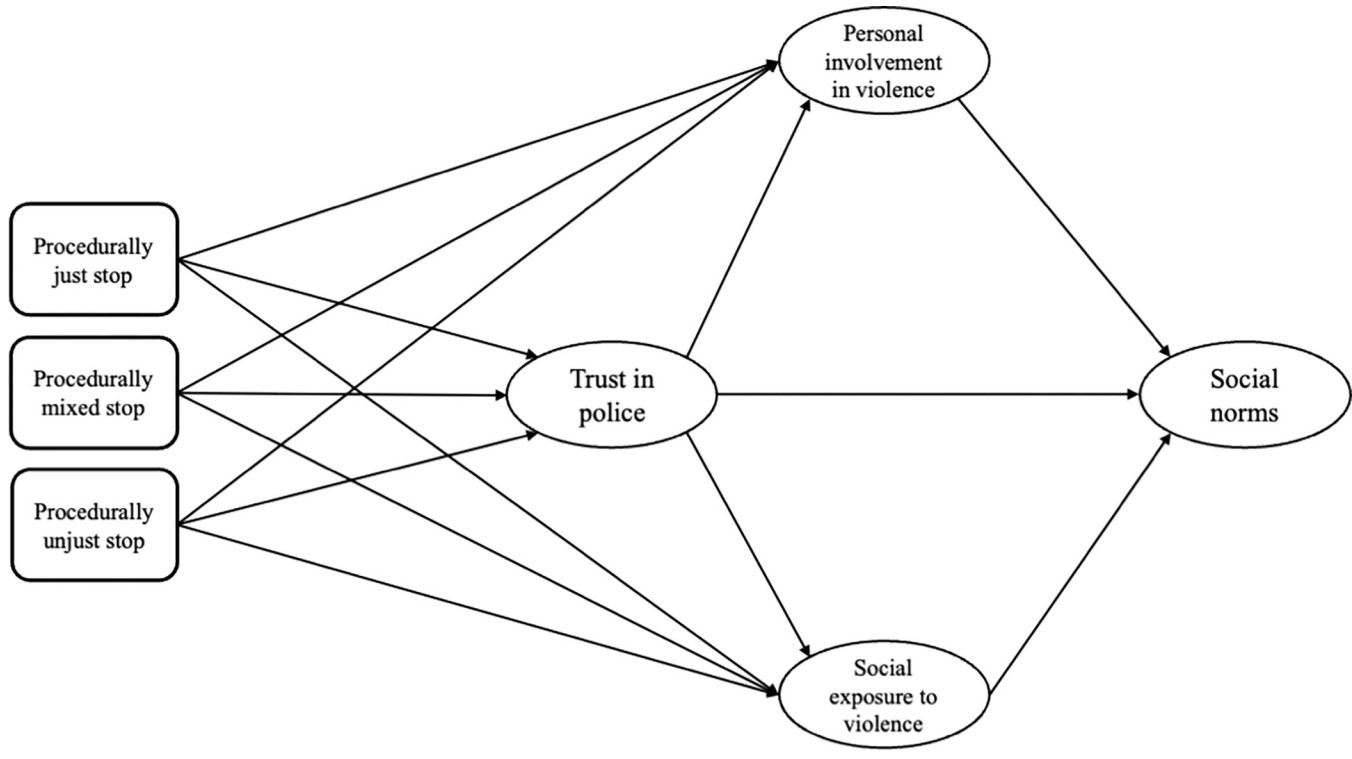

**Fig 1. Conceptual map.**

in crime reduction [3, 5] and the extent to which it is harmful to police-public relations [46]. S&S–and the wider category of 'police stops' [47]–is likely to be a key moment in the legal socialisation of a significant number of young people. Approximately 16% of the 67,997 S&Ss conducted in London from July to September 2020 involved young people aged 10–17 [6], a figure in line with previous years.

There are (at least) two reasons why S&S may shape trust in the police, and feed into normative attitudes and behaviours. These processes are bi-directional, and are typically envisaged as to at least some extent symmetrical. First, procedural justice during interactions with police indicates to people that the justice system police represent is founded on the right values, motivating reciprocal adherence to those values [48, 49]. Second, when people feel fairly treated by particular police officers, this strengthens their identification with the police and the group(s) they represent [50] and signals equality and even inclusive citizenship [51, 52] motivating, again, trust and adherence to group norms. *Unfair* treatment during S&S demonstrates to the individuals involved that police and the wider justice system do not operate in ways attuned to widely shared norms of fairness, diminishing the cognitive and affective 'pull' of those norms. Unfair treatment also weakens identification with the police and the superordinate social categories they represent–nation, state and/or community–reducing propensities to abide by group norms.

Yet, it is a widely replicated empirical finding that the effect of contact with officers is often *asymmetrical* [53]. While the experience of procedural injustice, particularly during police-initiated contacts, seems to have large negative effects on attitudes and opinions, the experience of procedural justice seems to have a smaller, albeit still significant, positive effect [54, 55]. This asymmetry persists among young people [56] and likely represents the fact that it is easier for the behaviour of authorities to undermine trust–which can happen in an instant–than to build trust, a process usually envisaged as time intensive [31].

**Gang exposure and involvement, violence and knife-carrying.** In addition to asking about S&S experiences, the MOPAC youth survey also asked participants about their involvement in a gang, carrying a knife or other weapon, and their social exposure (if any) to knife-carrying and violence/racist/hate-based material online. It is important to note that, when respondents report involvement and exposure, we do not assume that they are in a gang. We would also like to side-step the important discussion about the meaning and utility of the word 'gang' [57]. Instead, we treat exposure and involvement as indicating proximity to counter-normative attitudes and behaviour, and we link this to S&S, trust in the police, and gendered social norms.

To the extent that police target S&S activity appropriately, it might be expected that young people who have been stopped/searched will be more likely to be exposed to and perhaps involved in counter-normative activity (although it is worth recalling that most stops do not uncover any wrong-doing). However, we propose that it is not the mere fact of being stopped that will be important, but rather the extent of its perceived (un)fairness. Why might boys who have experienced *unjust* S&S have (on average) greater involvement in, and exposure to, knife carrying, violence, and hate-based content on social media? First, unfair S&S experience could damage the sense of safety and reassurance that flows from believing that the police and legal system can be trusted and are legitimate [19]. Brennan [24] explored whether people who believe that the police cannot provide security and lack the right intentions also feel the need to protect themselves. He found that a lack of trust in the police was an important predictor of knife carrying. An illegitimate justice system may also motivate people not only to provide for their own safety, but also take justice into their own hands [31]. Nivette [58], for example, found that people who saw the system as illegitimate also tended to be believe that violent *informal* norm enforcement, such as vigilantism, was acceptable.

Second, coercive policing activities that are experienced as unjust may weaken the moral force of the law [25, 27, 40, 59]. The argument here is that the erosion of legitimacy of legal authorities diminishes normative constraints on people's law-related behavior [37, 38]. This may be especially relevant for adolescents coming to learn about the law and legal authority [19]. Work on legal socialization suggests that teenagers and young adults can develop a 'healthy' relationship with the law and legal authorities based on a sense of mutual understanding and respect, or an 'unhealthy' relationship characterised by animosity and mistrust [18]. Where young people are located on this continuum depends in part upon the quantity and quality of their direct and indirect experiences of policing. A healthy relationship is associated with internalizing a sense of duty and responsibility to follow dominant standards of appropriate behaviour, independent of the context of rules and codes. An unhealthy relationship is associated with cynicism, disobedience, defiance and perhaps neutralization. As McLean & Wolfe [25, p.39] argue:

> "...perceptions of procedural injustice seem to make young people more likely to engage in techniques of neutralization . . . loosening "the moral bond of law" and allowing individuals to drift into criminal behaviour (Matza, 1964: 102) . . . Such behaviors appear to weaken . . . attachment to authority figures, thereby hindering the social control function such a bond normally exerts. By acting in manners inconsistent with normative expectations, police officers who are procedurally unfair may make it easier for adolescents to neutralise their otherwise legally compliant values . . . paving the way for more criminal behavior."

If unjust S&S experiences damage young people's feelings of obligation and responsibility to the law (and helps to push them away from conformity towards counter-normative behavior) then one indicator of this would be greater exposure to, and involvement in, violence and gang-related activity.

**Masculinity and gendered social norms.** The final part of the framework (right-hand side of Fig 1) captures the views of male adolescents on sexual harassment and controlling behaviour in intimate partner relations. Much has been written about the ways in which violence and self-protection, harassment, and controlling, aggressive and/or dominating behaviour express multiple forms of masculinity and masculinities [60–63]. The idea of hegemonic masculinity has gained considerable prominence. Men position themselves according to different meanings of hegemonic and non-hegemonic masculinity and adapt or distance themselves at different moments. This is not the place to delve in detail into the definitions, use and usefulness of this notion [60, 64, 65]; for the current study it is enough to identify *a priori* stereotypically masculine norms linked to the hegemony of men [64, 66] in society and the reproduction of gender inequality.

Also relevant here is Anderson's [67] 'code of the street', which focuses on oppositional masculine cultures based in part on violent, 'tough', identities that help to protect and to gain status and respect (when protection and status is denied by structural disadvantage). Interviewing men incarcerated in UK prisons for street violence, Brookman et al. [68] identified deeply gendered concerns in their accounts, including the importance of punishing disrespect, protecting themselves, self-reliance in conflicts, and having a fearful reputation. In ethnographic work set in Indianapolis Lauger [69, p.195] found that socially transmitted stories helped to frame and strengthen the meaning and use of violence as '. . .the expected response to the improper, illogical, and disrespectful action of peers' and to 'authenticate masculinity'. Equally, Palasinski & Riggs [23] found two main repertoires in people's accounts of knife-carrying: (1) attribution of blame to authorities for lack of protection and (2) masculinity and the desire for status.

On these accounts, certain forms of hegemonic masculinities lie behind practices like weapon-carrying and hate-based social media content, revolving around notions of self-protection and toughness, and the need for homo-social respect and competition. This may be a pertinent issue for the current study. Adolescence constitutes 'a critical window of development' for gender norms [70, 71]. As 'rules and standards that are understood by members of a group, and that guide and/or constrain social behavior without the force of laws' [72, p.152], norms constitute different pressures, within different social groups (e.g. societal norms, family norms, and norms specific to homosocial friendship groups), on what is and is not appropriate to do. *Gendered* social norms (a) define the expected behaviour of people who identify as either male or female [29] and (b) are generally constructed in the literature in a heteronormative, binary way [64].

There is a good deal of evidence that in early adolescence boys become more aware of social norms around gender, with some starting to reject feminine roles [73]. In middle adolescence there tends to be stronger relationships between male peers, as well as initial exploration of romantic and sexual relationships. In late adolescence there is more sexual activity, and established romantic and sexual relationships. Given how different elements of certain stereotypically masculine identities coalesce, e.g. the desire for male respect and self-protection may covary with sexual objectification and controlling females, it is reasonable to suppose that the particular masculinities being expressed in such practices have an 'elective infinity' (a 'force of mutual attraction involving the structure and contents of belief systems and the motives of their adherents', [74, p.309]) with certain attitudes towards gender relations and sexuality. When people gravitate to such attitudes and behaviours, power is realised, spaces and cultures are divided by gender, and "social spaces associated with men often [become] environments in which violence is rehearsed and reinforced" [75].

Capturing gendered normative attitudes, respondents to the MOPAC survey were asked whether they thought it was acceptable to stare, wolf-whistle and make sexual comments at people in the street, and whether they thought it was acceptable to tell their girlfriend or boyfriend not to hang out with certain people or check their phone and/or social media account. Like Burrell et al. [29] we consider "equality, respect and non-violence within intimate relationships" to be healthy, and dominant, social norms. Of concern here, therefore, are stereotypical gendered norms based on expectations of power and entitlement that serve to recreate inequality, render sexual harassment acceptable, and perhaps even help to trivialise violence against women [76, 77].

## Hypotheses

We argue that young people learn about the appropriate use of power from adults–such as police officers–who have power over them. While theories of legal socialisation would suggest that when police are involved this learning relates primarily to the law and legal authority, since power is unevenly distributed within many kinds of relationships, the learning process may extend into a wider set of situations and behaviours–here, personal and intimate relationships of a gendered nature.

The discussion above can be distilled into six hypotheses regarding the constructs set out in Fig 1, which guide the analysis that follows.

- H1: Experiences of procedural justice during S&S encounters will predict trust in the police, with procedurally unjust experiences having a stronger association than procedurally just experiences.

- H2: Experiences of procedural justice during S&S will be associated with exposure to and engagement in violence and gang-related behaviour.

- H3: Male adolescents with more trust in police will tend to exhibit fewer violence and gang-related behaviours.

- H4: Those with more trust in the police will tend to have more 'positive', i.e. more normative, attitudes towards gender relations and inter-personal relationships.

- H5: The association between procedural justice during S&S and violence and gang-related behaviour will be mediated by trust in police.

- H6: The association between procedural justice during S&S and normative attitudes will be mediated by both trust in police and violence and gang-related behaviour.

## Data and measures

**The survey.**  Analysis is based upon the Youth Survey 2018 conducted by MOPAC. The survey covered topics including feelings of safety at home and at school, experiences of crime victimisation, satisfaction with the police for young victims of crime, views and perceptions of the police, views of Safer School Officers, views and experiences of S&S, exposure to knife carrying and gangs, attitudes towards anti-knife crime campaigns, and experiences of online safety and sexual harassment (those in school years 10 and 11 only). For a full description of the survey see [78].

The survey was hosted online between 7 March 2018 and 8 May 2018. It was distributed to schools via the Metropolitan Police Service (MPS) Safer Schools Officers (dedicated police officers that work collaboratively with schools and educational establishments across the city). This method of distribution was selected to build upon existing partnership working between the MPS and schools. It also provided an ethical safeguard for young people by ensuring that consent from appropriate adults (teachers or parents) was obtained before participation. The survey was therefore not based on a random probability sample, and cannot be considered representative of the views of young people across London.

A total of 7,832 responses were received from young people aged 11 to 16, although in this paper we primarily use data only from those in school years 10 and 11 (n = 1,752) who identified as boys (n = 617). It is important to note that the survey contained a relatively high level of missing data. For example, 336 respondents (4% of the total sample) did not answer the question on S&S. Missingness was even greater on some of the demographic variables, presumably because some young people were reluctant to divulge personal and potentially identifying information. To help deal with this issue, we use multiple imputation (see below). However, 23% of those in years 10 and 11 did not answer the question on gender. We were forced to remove these cases from the main analysis, although they are included in supplementary analysis presented below.

**Response variables.**  Six items probed the views of respondents on the acceptability of certain behaviours concerning personal and intimate gendered relationships. These normative attitudes will to some degree be shaped by, and reflected in, gendered social norms about male and female roles. In all cases the response categories were: always OK, sometimes OK, never OK, and I don't know (set to missing). Due to the nature of the items, they were fielded to older children only (school years 10 and 11, aged 14 to 16). The items were:

1. Staring or wolf-whistling at someone you fancy as they walk past (*Stare*; 56% of boys said 'never OK').

2. Making sexual comments or jokes about what someone is wearing, or about how they look (*Joke*; 63% 'never OK').

3. Trying to dance with someone you fancy at a party, even if they say they don't want to dance with you (*Dance*; 72% 'never OK').

4. Checking your girlfriend or boyfriend's phone or social media account to see what they've been up to or who they've been talking to (*Check*; 49% 'never OK').

5. Telling your girlfriend or boyfriend not to hang out with certain friends because you don't like them (*Friends*; 61% 'never OK').

6. Saying nasty things to your girlfriend or boyfriend during an argument (*Insult*; 73% 'never OK').

There was thus general agreement that these behaviours were inappropriate. Only in the case of *Check* (checking your girlfriend's or boyfriend's phone or social media account to see what they've been up to or who they've been talking to) did the percentage saying a behaviour was 'never OK' fall below 50 per cent. That said, there was variation: some did think it was OK to engage in these behaviours. In addition, note that *Stare*, *Joke* and *Dance* are generic, while *Check*, *Friends* and *Insult* refer explicitly to relationships (boyfriend/girlfriend). Our baseline assumption was that responses to these six items would vary across two underlying dimensions. Confirmatory factor analysis (in the package Mplus 8, treating the observed indicators as categorical and using full information maximum likelihood estimation to deal with missing values) supported this reasoning. A two-factor model specified as above, with no cross loadings fitted the data well (Chi$^2$(6) = 9.9, p = .13; RMSEA = .04, RMSEA$_{90\%}$ = [< .0005 .07]; SRMR = .02; CFI = .99; TLI = = .99). By contrast, a one-factor model that grouped all observed indicators together (Chi$^2$(9) = 108.5, p = .14; RMSEA = .14, RMSEA$_{90\%}$ = [.12 .17]; SRMR = .07; CFI = .90; TLI = .84) had a worse fit, as confirmed by use of the Mplus DIFFTEST option [79]. The correlation between the two factors was high (r = .90), but on the basis of the fit statistics we proceed with this solution.

We therefore have two response variables, representing views on norms concerning behaviours that might be considered 'generically' wrong, relating to sexual harassment in public space (labelled *Norms 1* for brevity), and those that are aggressive and potentially controlling within intimate partner relationships (*Norms 2*). In both cases higher values correspond to less acceptance (or stronger rejection) of the behaviours concerned.

**Explanatory variables.**    There are three main groups of explanatory variables. First, experience of S&S: respondents were asked whether they had 'ever' been stopped and searched by police. Overall, some 21 per cent of boys in school years 10 and 11 answered yes; like Geller [80] in the US, we find that police stops are an important aspect of growing up in a city like London (see also [81]). Those who had been stopped and searched were asked three follow up questions about their most recent experience (with yes/no responses): whether the police had been polite (35% said yes); treated them with respect (30% said yes); and explained why they had been stopped and searched (41% said yes). To operationalise S&S experiences we created three dummy variables: stopped and 'full' procedural justice (answered yes to all three follow up questions); stopped and 'mixed' procedural justice (answered yes to at least one follow up question); stopped and 'no' procedural justice (answered no to all follow up questions). The reference category was therefore 'had not been stopped and searched'.

Second, respondents' own weapon- and gang-related behaviours, and social exposure to gangs and violent material online were also measured. *Personal involvement in gang-related activity* was measured by asking participants whether they had ever been part of a gang, carried a knife, felt pressured to carry a knife or carried any other weapon other than a knife. While carrying a knife or other weapon is (in this context) probably illegal, doing so does not necessarily imply intention to use it; many young people who carry knives do so for their own

protection [82]. Similarly, gang membership can be a way of seeking safety and a positive sense of self [83], rather than an active choice to commit crime or anti-social behaviour. We interpret the answers young people gave to these items as representing, broadly speaking, their responses to particular social and environmental cues [82]–that concerning us here being distrust in the police and the action alternatives that flow from this–rather than commitment to offending behaviour *per se*.

*Social exposure* to gangs and violence was measured by asking respondents whether they knew anyone who carried a knife, and whether they had seen in the last year any content on the internet or social media that showed violence (e.g., videos showing fights or use of weapon), gang-related content (e.g., pictures/videos showing gang activity), or racist content or content that promotes hate or discrimination.

For both groups of questions, the response alternatives were yes and no (don't know and don't want to say responses were coded to missing). The prevalence of 'no' responses for personal involvement was much higher than for social exposure. To examine the association between these measures, we used confirmatory factor analysis in Mplus 8, with all items set to ordinal and Full Information Maximum Likelihood estimation to deal with missing values. A two-factor solution with no cross-loadings showed the best fit ($Chi^2(19) = 89.3$, p<0.0005; RMSEA = 0.08, $RMSEA_{90\%}$ = [.06 .10]; SRMR = 0.11; CFI = 0.98; TLI = 0.97) with the personal involvement and social exposure items loading on distinct factors (which were only moderately strongly correlated, $r$ = .55). For both factors, higher values correspond to *more* involvement or *greater* exposure.

The final explanatory variable was trust in the police. This was a scale created from four survey items probing respondents' perceptions of the helpfulness and friendliness of police, whether police treat young people the same as adults, whether police treat everyone fairly, and whether police listen to the concerns of young people. Factor analysis, as before, demonstrated all these items loaded onto one underlying construct, with the one factor solution showing good fit ($Chi^2(2) = 4.8$, p = .09; RMSEA = .05, $RMSEA_{90\%}$ = [< .0005 .11]; SRMR = .01; CFI = .99; TLI = = .99). Note that while we label this measure trust in the police (for brevity), it relates primarily to general perceptions of police fairness.

**Control variables.** A number of control variables were also used: ethnicity (as a White/non-White indicator due to sample size issues); age (although almost all respondents were aged either 15 or 16); and victimisation, measured with a simple binary variable where the response category was no victimisation experience. To capture some of the likely variation across different parts of the capital, standard errors were assumed to be clustered by the respondents' borough of residence. Table 1 shows the distribution of key categorical variables for the boys in Years 10 and 11 used in the main analysis.

## Results

We used structural equation modelling in Mplus 8 to address the hypotheses laid out above (again using full information maximum likelihood estimation and using a further 24 questions from the survey to inform the multiple imputation of missing values). Fig 2 shows results from the model tested with age, ethnicity, the procedural justice of police during S&S encounters, and victimisation as exogenous variables. In this model, trust was regressed on the S&S measures, the gang- and violence-related latent constructs were regressed on trust and the S&S measures, and the social norms variables were regressed on all other measures. For visual ease only the significant pathways from police S&S and victimisation onwards are depicted by Fig 2 (the full output is available from the authors). Note that the approximate fit statistics are marginal, compared to accepted cut-offs. This is due to the requirements for multiple imputation

**Table 1. Descriptive statistics.**

| | % | n |
|---|---|---|
| **Age** | | |
| 14 | *23* | 142 |
| 15 | *49* | 299 |
| 16 | *28* | 170 |
| **Ethnic group** | | |
| Asian | *25* | 149 |
| Black | *19* | 111 |
| Mixed | *8* | 50 |
| White | *38* | 224 |
| Other | *10* | 61 |
| **Ever stopped and searched** | | |
| Yes | *21* | 127 |
| No | *76* | 469 |
| Don't know | *3* | 19 |
| Did not answer | *<1* | 2 |
| **Most recent stop experience (those stopped only)** | | |
| Procedurally just | *18* | 23 |
| Mixed | *46* | 59 |
| Procedurally unjust | *35* | 45 |
| **Victim of crime in last 12 months** | | |
| Yes | *16* | 99 |
| No | *70* | 431 |
| Don't know | *8* | 50 |
| Don't want to say | *6* | 36 |

Boys, Years 10 and 11 only.

and clustered standard errors (for example observed indicators were set to continuous). We estimated a second SEM, with no multiple imputation, no clustered standard errors, and observed indicators set to ordinal. Here, model fit was better (e.g. RMSEA = .04; CFI = .96; TLI = .95; SRMR = .07). Significant/non-significant associations between variables almost exactly replicate those shown in Fig 2, but effect sizes are generally larger. We proceed with the model shown in Fig 2 as the more conservative option.

We find, first, that the partial association between trust in the police and S&S experiences was primarily negative (or more accurately asymmetrical). Compared with those who had not been stopped and searched, those who experienced a 'fully' fair stop did not trust the police more. Conversely, those who experienced stops with 'mixed' procedural justice or no procedural justice were significantly less likely to trust the police. These findings mirror those from other studies which have found an asymmetrical association between police-initiated contact and trust.

Second, the gang- and violence-related latent constructs showed negative conditional associations with S&S and victimisation. The more procedurally unjust the S&S was, the stronger negative relationship it had with *personal involvement* (procedurally just: ß = .01, p = .68; mixed: ß = .23, p < .0005; unjust: ß = .26, p < .0005). In comparison, *social exposure* had a partial *positive* association with the procedurally just (ß = .10, p < .0005) stops only. This latter finding is the opposite of what might be expected, stands in contrast to the pattern of associations elsewhere in the model, and may simply be a Type I error. Being victimised showed a

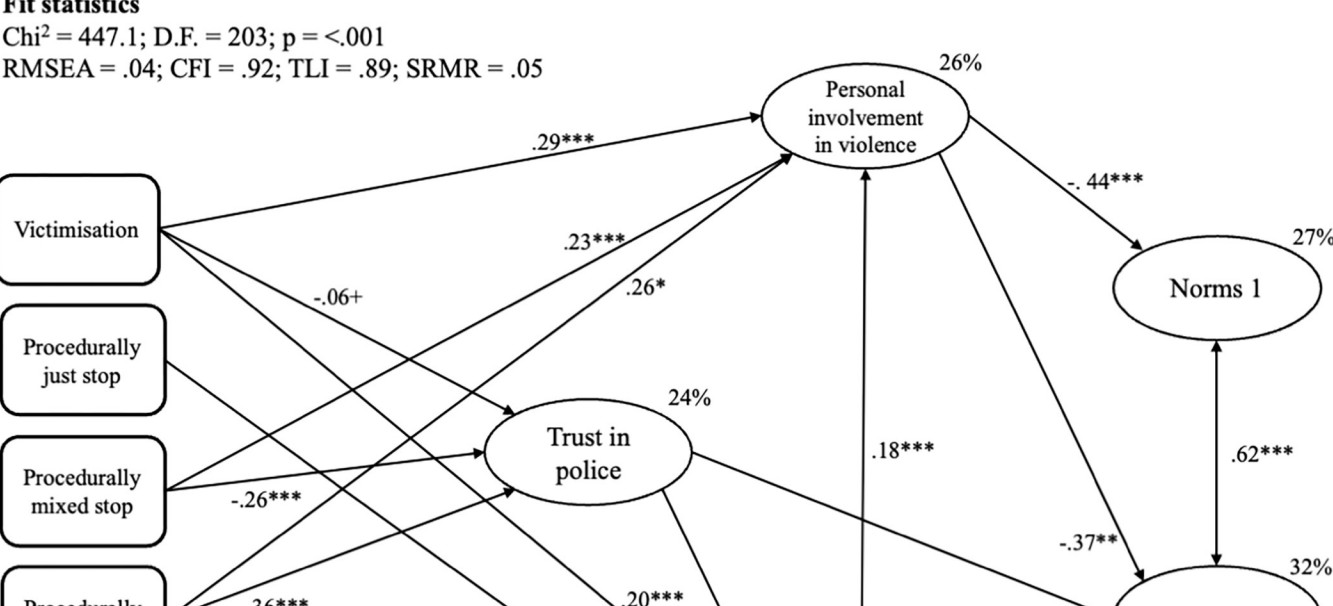

**Fit statistics**
Chi$^2$ = 447.1; D.F. = 203; p = <.001
RMSEA = .04; CFI = .92; TLI = .89; SRMR = .05

**Notes**
All endogenous variables also regressed on ethnic
group and age – paths not shown for visual ease.
Non-significant paths (p>.1) also excluded
*** p <.001; ** p<.01; * p<.05 + p<.1

**Fig 2. Results from a structural equation model predicting attitudes towards gendered social norms (boys only).**

relatively strong positive partial association with both personal involvement and social exposure (i.e. recent victims of crime were more likely to be involved in or exposed to violence).

Third, trust in police was also associated with personal involvement, but appeared unrelated to social exposure. In addition, trust in police was positively correlated with attitudes towards social norms. Controlling for the other variables in the model, boys who trusted the police more were less likely to think that aggressive and potentially controlling within intimate partner relationships (Norms 2) were 'OK' (ß = .15, p = .06); the relationship between trust and sexual harassment in public space (Norms 1) was similar in magnitude but not statistically significant (ß = .10, p = .14). We also find that those personally involved in or socially exposed to violence tended to be more likely to think norm-transgressing acts acceptable (the only *non-significant* path here was that between social exposure and Norms 1, ß = .08, p = .21). Across all the explanatory variables, personal connection to violence had the strongest conditional relationship with both groups of norms.

To consider the notion that S&S experiences may shape the normative attitudes and behaviours of young people via trust in the police, indirect effects were also assessed. Trust in police mediated the impact of 'mixed' (IE = .06, p = < .0005) and unjust stops (IE = .08, p = < .0005) on social exposure. There is some evidence that trust also channelled the statistical effect of mixed (IE = -.04, p = .09) and unjust (IE = -.06, p = .06) stops to Norms 2. Personal involvement channelled the impact of both procedurally mixed and unjust stops on Norms 1 and 2

(e.g. the indirect effect of unjust stops on Norms 2 via personal involvement was IE = -.09, p = 03), as did personal involvement (e.g. the indirect effect of unjust stops, via personal involvement, on Norms was IE = -.11, p = .06; and on Norms 2 it was IE = -0.09, p = .03). These results provide further evidence that the influence of previous experiences with the police on boy's normative attitudes is transmitted by the intermediate variables specified in our model–trust in police and personal involvement in gang-related activity.

Finally, note that the *total* statistical effects of trust in the police on Norms 1 (ß = .04, p = .06) and in particular Norms 2 (ß = .21, p < .01) were both in the expected directions. Similarly, the total effect of just stops on Norms 1 (ß = -.05, p = .44) and Norms 2 (ß = -.08, p = .18) was relatively small and non-significant, while the total effect of unjust stops was much largely and strongly significant in both cases (Norms 1 ß = -.18, p = < .0005; Norms 2 ß = -.28, p = < .0005).

## Additional analysis

We have concentrated thus far on the views and experiences of boys, who are more likely to be stop/searched by police and, it might be assumed, more likely to learn gender-related messages from the behaviour of officers. However, all young people are exposed to this form of police activity, and are moreover equally exposed to norms of hegemonic masculinity. As a check, we therefore estimated the model shown above for the full year 10 and 11 sample, including girls, non-binary people and those who did not answer the gender question. As before, we used full information maximum likelihood estimation and a further 24 questions from the survey to inform the multiple imputation of missing values.

Results from this model are shown in Fig 3, and they correspond very closely with those in the boys-only model shown in Fig 2. In this model, however, trust in police is significantly and positively associated with both Norms 1 and Norms 2. The indirect statistical effects of procedurally mixed and unfair stops on Norms 1 and Norms 2 were also significant and in the expected directions. For example, the total indirect effect of procedurally unjust stops on Norms 1 was IE = -.16, p < .0005 (although note that the total indirect effect of procedurally just stops on Norms 1 was also negative, albeit much smaller; IE = -.03, p = .03). In this model, trust in the police also channelled the statistical effects from negative stop and search experiences to normative attitudes; for example, the indirect statistical effects of mixed and procedurally unjust stops, via trust, to Norms 2 were both significant (IE = -.04, p = < .01 and IE = -.05, p = < .0005, respectively).

## Discussion

We found significant but qualified support for our hypotheses. Experiences of procedural (in) justice during S&S encounters predicted trust in the police (H1) and exposure to gangs and violence (H2). We found partial support for H3, because trust in police was only associated with social exposure to such behaviours, but not personal connection. H4 stated that male adolescents with greater trust in police would tend to have more 'positive' norms regarding gendered personal and intimate relationships. Again, we found partial support this hypothesis; while direct effects were marginal in size and statistical significance, the total statistical effect was stronger, particularly in relation to Norms 2 (and it was stronger, and more direct, in the 'full sample' model). Finally, the hypotheses regarding mediated pathways received partial support. Trust in police mediated the statistical association between S&S experiences and social exposure to, but not personal involvement in, gangs and violence (H5 was partially supported). Similarly, both involvement in and social exposure to violence mediated the statistical effect of procedural justice during S&S on normative attitudes; the mediating role of trust in the police was weaker, but could still be identified (H6).

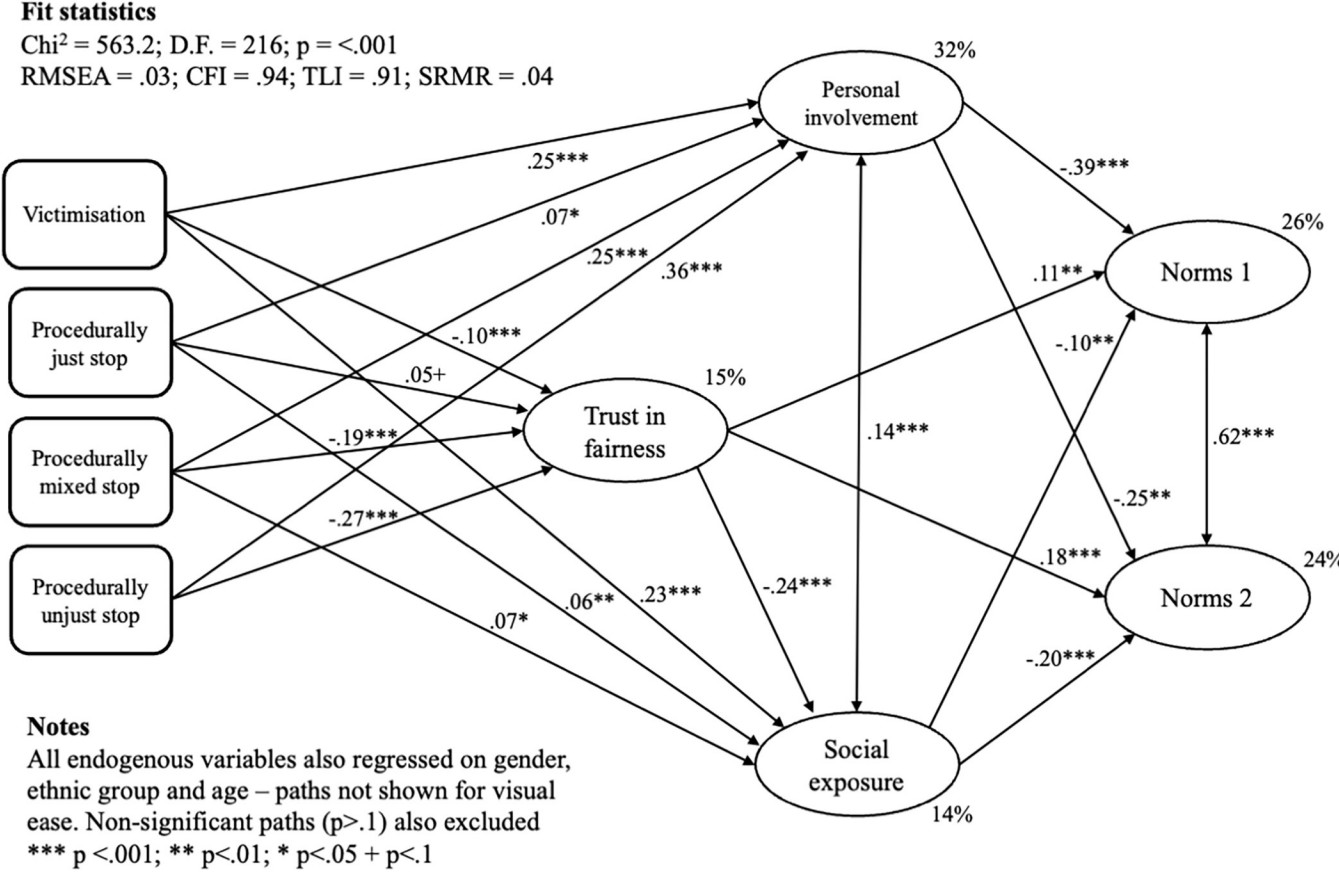

**Fit statistics**

Chi$^2$ = 563.2; D.F. = 216; p = <.001
RMSEA = .03; CFI = .94; TLI = .91; SRMR = .04

**Notes**

All endogenous variables also regressed on gender, ethnic group and age – paths not shown for visual ease. Non-significant paths (p>.1) also excluded
*** p <.001; ** p<.01; * p<.05 + p<.1

**Fig 3. Results from a structural equation model predicting attitudes towards gendered social norms (all respondents).**

In sum, the findings presented here support the idea that regulatory encounters between police and young people may be important moments in the formation of attitudes toward social norms. Compared with those who had not been stop/searched, *or who had an experience that they found procedurally just*, boys who had experienced procedurally unjust stop/searches tended to trust the police less, were more likely to be involved in gang- and violence-related behaviours, and had less positive attitudes towards behaviours relating to sexuality and intimate partner relationships. This is consistent with the idea that policing experienced as procedurally unjust is associated with unhealthy attitudes towards not only violence and aggression, but also sexual objectification and controlling behaviour within relationships (all linked in some degree to stereotypically masculine norms regarding acting tough, being sexually aggressive, and imposing control on intimate partners).

Why might this pattern of correlations be plausible? It may be that when police officers treat people with procedural justice, they communicate that the values embedded in the law are appropriate *and* that legal authorities behave in accordance with these values. The idea that people with power should behave fairly is widely and deeply held. Research has shown, for example, that children and young people value procedural justice in interactions with both parents and teachers [84–86]—in other words, in their relationships with authority figures that in the vast majority of cases precede their encounters with police officers. Most young people therefore 'go into' encounters with police expecting and wanting to be treated fairly; when they are, they conclude that the values police represent are correct and aligned with their own.

In such cases they will at least, and all else equal, be no *less* likely to conclude that the existing normative legal order is dependable, and feel no *more* need to engage in non-normative behaviours (either by seeking out materials online or by getting involved in informal norm enforcement themselves such as carrying a knife) than if the encounter had never occurred.

S&S encounters may thus prompt or motivate an orientation to non-normative behaviours that is in line with both the law and wider social attitudes. Importantly, though, we found that the negative statistical effects of procedurally unfair treatment seem to be stronger than any positive effects of encounters that were judged as procedurally just. It could be easier for police S&S activity to undermine young people's commitment to certain normative attitudes than to strengthen it. In particular, male adolescents who receive an unjust 'lesson' from police may be more likely to turn to hegemonic masculine norms linked to violence, the need to assert invulnerability, and the desire to be seen as 'street smart'. At the same time, unfair teachable moments during S&S may promote a sense that the abuse of interpersonal power is acceptable. These things, together, may promote a sense that it is OK to abuse, objectify and control females.

While it is important to remember that young people's normative beliefs about relationships and how to behave in intimate partner relations will largely be shaped in non-policing contexts (family, home, schools, peer-groups), it therefore seems plausible that police behaviour can also have some effect. There is an important potential implication for policy here. One might view stop/searches of young people as 'one-off' encounters that have little lasting effect on the individuals concerned. Yet, our findings suggest these are interactions that, when poorly handled, may *encourage* young people to seek out or get involved in gang-related behaviours, and shape their attitudes towards what is and is not appropriate behaviour (and, to the extent that attitudes correlate with behaviours, perhaps their future actions too). S&S may therefore be seen as a way of influencing the normative attitudes of those exposed to it, and the effects of a stop may ripple forward into seemingly disconnected attitude and perhaps behaviours. It may be part of the 'tragedy' of the power that this seems to be an asymmetrical, negative process, which might even be criminogenic in nature.

We should note the limitations of this paper. First, the cross-sectional nature of the data does not allow us to establish causal order. Even though the hypothesised pathways in the SEM model were supported by theory and earlier empirical evidence (often longitudinal studies), the conditional correlations in the model have to be interpreted with caution. In particular, it may be that (a) police target people who have counter-normative attitudes and (b) there is something about both the dynamic of the encounter and the ways in which the young people involved 'selectively' view procedural justice that could be shaped by their counter-normative attitudes (in other words they are primed in some way to experience policing as unfair). One defence against such charges is, of course, that even by the most favourable estimates the majority of people stopped by the police are found to have done nothing wrong. It also seems implausible that officers are, at scale, somehow able to pick out to stop and search London teenagers with counter-normative gendered attitudes; and recall that it was the experience of procedural *injustice* was crucial in our findings, not the mere fact of being stopped. Moreover, the most common grounds for S&S is possession of a controlled substance, usually small amounts of cannabis, and there is no particular reason to suggest that recreational drug users have less normative attitudes towards personal relationships than anyone else. Nevertheless, it will be important for future longitudinal studies to properly test to pathways we have proposed here.

Second, and as a general limitation of this type of research, it is impossible to determine whether those respondents who claimed to experience a procedurally unjust S&S were actually treated unfairly [87]. It has been shown that previous attitudes toward the police can heavily

affect how one assesses a certain encounter [88], making it possible that police actions are judged with undeserved prejudice. Third, due to data limitations, and despite controlling for age, ethnicity and victimisation, we could not account for other potentially influential covariates, such as individual and neighbourhood disadvantage or parental views of the legal system [56, 89]. Similarly, we could not consider other important aspects of the stop and search encounter itself, such as the gender of the police officers involved and the presence or absence of witnesses or bystanders. Future research should address all these issues.

## Conclusion

One of the central questions for policing over the next decade will be how to foster better relationships with young people. We begin this process at a time when concerns about serious offending are prompting shifts to more enforcement-lead tactics (among other things), and police community relations have been put under some strain as a result of fallout from the Black Lives Matter movement (and the historic wrongs it seeks to challenge), Covid-19, and a range of other factors. It will be vital to find ways to address this challenge that attend to the 'natural', quotidian, encounter between police and young people. We have concentrated here on one of the most fraught and problematic of such encounters, S&S. It is perhaps unsurprising that we find 'asymmetry'. Yet, other forms of contact between police and public have been found to be more 'symmetrical' [55], offering greater possibility of police affecting positive influence on the normative attitudes of young people. It may be that these more positive interactions can foster healthier normative attitudes towards gendered relationships than those we have focussed on here.

Again, more research is needed. But if our findings here were to be supported by future work, this would have significant implications for how we think about the effect of police activity on young people's relationships not only with the police, but also with each other. To our minds, this issue is important enough that it should be seen as an organisational challenge for police forces and Police and Crime Commissioners, and be embedded within training, standards of practice and discussions as to what a good police force looks like. Yet, shifting the way police officers think about their encounters with young people is unlikely to be easy, and may require significant investment in training and other activity. For example, there is existing research that sets out officer bias and stigmatisation of individuals with mental health, along with specific training to address this [90, 91]. One question may well be to ask whether a similar stigma exists towards youth, particularly those that officers encounter on a regular basis—their 'usual suspects' [92]—and whether greater efforts to enhance the 'youth encounter' are also required.

## Supporting information

**S1 File.**
(ZIP)

## Author Contributions

**Conceptualization:** Ben Bradford, Paul Dawson.

**Formal analysis:** Krisztián Pósch.

**Methodology:** Ben Bradford, Krisztián Pósch.

**Supervision:** Ben Bradford.

**Writing – original draft:** Ben Bradford, Jonathan Jackson, Paul Dawson.

**Writing – review & editing:** Ben Bradford, Jonathan Jackson, Paul Dawson.

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
