## [Decision Letter · Decision Letter 0]

23 Aug 2022

PONE-D-22-18323A street corner education: Stop and search, trust, and gendered social normsPLOS ONE

Dear Dr. Bradford,

Thank you for submitting your manuscript to PLOS ONE. After careful consideration, we feel that it has merit but does not fully meet PLOS ONE’s publication criteria as it currently stands. Therefore, we invite you to submit a revised version of the manuscript that addresses the points raised during the review process.

We look forward to receiving your revised manuscript.

Kind regards,

Guangyu Tong

Academic Editor

PLOS ONE

Journal Requirements:

Additional Editor Comments:

Please address all the reviewer comments accordingly. We look forward to receiving your revision.

Reviewers' comments:

Reviewer's Responses to Questions

**Comments to the Author**

1. Is the manuscript technically sound, and do the data support the conclusions?

Reviewer #1: Partly

Reviewer #2: Yes

2. Has the statistical analysis been performed appropriately and rigorously? 

Reviewer #1: I Don't Know

Reviewer #2: Yes

3. Have the authors made all data underlying the findings in their manuscript fully available?

Reviewer #1: No

Reviewer #2: No

4. Is the manuscript presented in an intelligible fashion and written in standard English?

Reviewer #1: Yes

Reviewer #2: Yes

5. Review Comments to the Author

Reviewer #1: Review of Manuscript PONE-D-22-18323

“A Street Corner Education: Stop and search, trust, and gendered social norms”

This paper has the potential to make a significant contribution to the literature on police contact among youth, and its role in legal socialization (and adolescent socialization more broadly). The focus specifically on gendered harassment and relationship dynamics is particularly innovative – it builds on the literature on police contact and masculinity, but identifies measurable outcomes closely tied to behavior. The paper is limited by its use of cross-sectional data, but the correlational relationships observed in the model pave a clear pathway for future research and data collection to assess causal effects.

I offer several suggestions for strengthening the paper:

• First, although the data for this paper were collected in the UK, there is some work based on youth and young adults in US cities that are relevant for the paper – specifically, Monica Bell’s (2017, Yale Law Journal) work on procedural injustice is critical to cite. In addition, Amanda Geller (2021, American Journal of Public Health) provides population level prevalence of police contact among adolescents, and Geller and Fagan (2019, Russell Sage Foundation journal) provide an examination of police contact and legal socialization. Additional work based on the Fragile Families and Child Wellbeing Study may also be of interest.

• The paper needs a table of descriptive statistics – a reader should be able to look at this “Table 1” and easily understand the prevalence and distribution of the key treatment, outcome, and covariates.

• The outcome measures are interesting and (particularly in the questions on gendered harassment) innovative, but vulnerable to social desirability bias (and potentially in either direction – if boys are willing to engage in harassing behavior while out with peers but unwilling to report willingness to engage in this behavior to adults… or if they overreport such willingness in attempts to look tough.). What is the risk that this social desirability is driving your results? What direction would the over/underreporting need to be in, and among whom? Is this plausible?

• The structural equation models seem reasonable, though I’m not an expert on this particular modeling method – I imagine other reviewers may be better able to comment on the specifics. That said, I have some concerns about the inferences the authors are making – they are careful to caveat their findings and note that their ability to make causal inferences from cross-sectional data is limited. Nonetheless, the discussion, while well written, clearly suggests causal effects. The authors should be a bit more careful not to overstate their conclusions.

I think this paper has a lot of potential and wish the authors luck with revisions.

Reviewer #2: The authors present a unique study examining the impact of police interactions on trust in police, involvement/exposure to gang-like activities, and how all of these experiences and perceptions impact stances on gendered norms. The authors find some support for their arguments in boys-only models (e.g., that individuals personally involved in/exposed to violence tended to find transgressions of gendered norms more acceptable & boys who trust police were less likely think aggressive/controlling relationships are “OK”). The authors also offer a test examining girls and respondents who didn’t identify their gender and report that findings were similar. This review presents some suggestions and lingering questions I had.

• At some points throughout the manuscript, it seems that the findings from Epp et al. (2014) [Pulled Over: How Police Stops Define Race and Citizenship] may have some implications worth noting—though Epp et al.’s research is based in the US. Specifically, the authors look at traffic stops: the type stops the officers make, how the officers interact with the person stopped, the race of the driver stopped. When stops are common, repeated, “unjust,” the process is seen as unfair and conveys messages about equality and citizenship.

o This Epp et al. research came to mind regarding the sentences on line 58 & 59 (page 3) and “Unfair treatment…” on line 221 (page 9)

• Line 86 (page 4) – you say, “mainstream norms of respect and equality between males and females”, but are those the mainstream gender norms? Could power differentials/“traditional” male/female social role divides be understood as more “mainstream”?

• I agree that masculinity is an important concept here, but you do not bring up masculinity until page 12. I think it may be worth mentioning earlier. It seems to take a while for the gendered component of your argument to be brought to light though it is central to your study. I would recommend adding more detail at earlier points. For one example, you could maybe add a little more detail when you say: “internalise unhealthy gendered norms” (line 191, page 8).

o Do you think the gender of the involved officers matter? What about if there are/are not any witnesses of the S&S experience?

• Is the additional analysis inclusive of all boys, girls, and those who didn’t answer since the sample size is 1,666? At first it sounded like you were only going to look at girls and those who didn’t answer the gender question here.

o I am also wondering about your inclusion of cases where individuals didn’t answer the gender question. Does the inclusion of respondents who didn’t answer the gender question make the results less comparable to the main analysis? Why not keep the additional analysis to just girls while the main models look at boys?

o You say the findings from the additional analysis align closely with the full models, but I’m wondering about possible bias in the results? (--especially if boys were still included here too). And you said 23% of 10 and 11 school year students didn’t answer (about 383 students if my calculation is correct), which is a pretty large percentage.

• What was the ethnic breakdown of this survey? You control for it, but I’d be interested in seeing what the distribution looks like. Also, what does BME stand for?

Suggested grammatic and structural edits:

• MOPAC first appears in line 94 (page 5) and should be written out long form at that point, then abbreviated after that.

• The two sentences spanning lines 101-103 (page 5) “It is important to note… our analysis is therefore speculative.”) can probably be removed here and just be included in the limitations section of the Discussion.

• The sentence on lines 303-305 sounds a bit awkward (page 12) (“…Lauger (2014) found that…”)

• The sentence spanning line 333-336 (p. 13) (“Capturing gendered normative attitudes…) seems a bit out of place; it can probably wait until the Methods section when you talk about the data and measures used in greater detail.

6. PLOS authors have the option to publish the peer review history of their article (what does this mean?). If published, this will include your full peer review and any attached files.

Reviewer #1: No

Reviewer #2: No

---

## [Author Response · Author response to Decision Letter 0]

11 Oct 2022

Reviewer #1

• First, although the data for this paper were collected in the UK, there is some work based on youth and young adults in US cities that are relevant for the paper – specifically, Monica Bell’s (2017, Yale Law Journal) work on procedural injustice is critical to cite. In addition, Amanda Geller (2021, American Journal of Public Health) provides population level prevalence of police contact among adolescents, and Geller and Fagan (2019, Russell Sage Foundation journal) provide an examination of police contact and legal socialization. Additional work based on the Fragile Families and Child Wellbeing Study may also be of interest.

We thank the reviewer for pointing out these references, and we had added them to the text.

• The paper needs a table of descriptive statistics – a reader should be able to look at this “Table 1” and easily understand the prevalence and distribution of the key treatment, outcome, and covariates.

Now added

• The outcome measures are interesting and (particularly in the questions on gendered harassment) innovative, but vulnerable to social desirability bias (and potentially in either direction – if boys are willing to engage in harassing behavior while out with peers but unwilling to report willingness to engage in this behavior to adults… or if they overreport such willingness in attempts to look tough.). What is the risk that this social desirability is driving your results? What direction would the over/underreporting need to be in, and among whom? Is this plausible?

The reviewer is of course right that social desirability may well be an issue here. One defence is that significant number of respondents did in fact indicate that it was at least sometimes OK to engage in the behaviours used to construct the social norms scales – there was variation in views. But another point is that social desirability is, plainly, related to what respondents perceive to be dominant social norms. If those high in trust in police, for example, were more motivated to provide what they felt was the ‘right’ answer than those who were lower in trust, this does not undermine our argument – indeed in a sense it is our argument. The issue then becomes the extent to which these expressed opinions are related to more deeply held attitudes, and the extent to which all this guides behaviour. We hope that future work will address these important questions.

• The structural equation models seem reasonable, though I’m not an expert on this particular modeling method – I imagine other reviewers may be better able to comment on the specifics. That said, I have some concerns about the inferences the authors are making – they are careful to caveat their findings and note that their ability to make causal inferences from cross-sectional data is limited. Nonetheless, the discussion, while well written, clearly suggests causal effects. The authors should be a bit more careful not to overstate their conclusions.

We have been through the discussion to ensure more careful language is used throughout.

Reviewer #2

• At some points throughout the manuscript, it seems that the findings from Epp et al. (2014) [Pulled Over: How Police Stops Define Race and Citizenship] may have some implications worth noting—though Epp et al.’s research is based in the US. Specifically, the authors look at traffic stops: the type stops the officers make, how the officers interact with the person stopped, the race of the driver stopped. When stops are common, repeated, “unjust,” the process is seen as unfair and conveys messages about equality and citizenship.

o This Epp et al. research came to mind regarding the sentences on line 58 & 59 (page 3) and “Unfair treatment…” on line 221 (page 9)

We have added this reference

• Line 86 (page 4) – you say, “mainstream norms of respect and equality between males and females”, but are those the mainstream gender norms? Could power differentials/“traditional” male/female social role divides be understood as more “mainstream”?

We are grateful for this point. We have removed the note about these being “mainstream” norms. This claim is both unnecessary to our argument and begs questions.

• I agree that masculinity is an important concept here, but you do not bring up masculinity until page 12. I think it may be worth mentioning earlier. It seems to take a while for the gendered component of your argument to be brought to light though it is central to your study. I would recommend adding more detail at earlier points. For one example, you could maybe add a little more detail when you say: “internalise unhealthy gendered norms” (line 191, page 8).

We have added references to gender at the place the reviewer suggests, and also earlier in the paper (e.g. line 59 [note that line numbers will have changed with editing). In particular, we have added the following to the introduction:

“Gendered social norms define the practices that are expected of males and females, typically in binary, heteronormative ways (Burrell et al., 2019). Starting from the premise that certain stereotypical forms of hegemonic masculinities coalesce around notions of toughness, competition and self-protection, as well as sexual objectification in public space and control within intimate-relationships, we consider the idea that gendered norms (based on gendered dominance that render it acceptable to sexually harass in public space and to control romantic partners) could be activated and/or strengthened when normative constraints are loosened by the (subjective) experience of procedural injustice.”

o Do you think the gender of the involved officers matter? What about if there are/are not any witnesses of the S&S experience?

These are important points, but ones we unfortunately cannot cover in the current paper. We have added a sentence to the limitations section to make the point that ideally these issues would be covered, and should be in future work.

• Is the additional analysis inclusive of all boys, girls, and those who didn’t answer since the sample size is 1,666? At first it sounded like you were only going to look at girls and those who didn’t answer the gender question here.

The additional analysis contains the full Year 10/11 sample, i.e. boys, girls, those with another gender identity and those who did not answer the question (note also tht we have corrected the sample size here to 1,752). This seems preferable to simply focussing on girls, primarily because it allows inclusion of the relatively large number of respondents who did not report their gender and whom, therefore, could conceivably be different to other respondents in important ways (e.g. trust). It may be important that in this second model the paths from trust in the police to Norms1 and Norms2 strengthen, and both are significant at the .05 level

o I am also wondering about your inclusion of cases where individuals didn’t answer the gender question. Does the inclusion of respondents who didn’t answer the gender question make the results less comparable to the main analysis? Why not keep the additional analysis to just girls while the main models look at boys?

As noted above, it seems more important to include the gender ‘non-responders’ than simply focus on girls. We could of course introduce a third model that looks only those who identified as girls, but this only a small number with stop and search experiences.

o You say the findings from the additional analysis align closely with the full models, but I’m wondering about possible bias in the results? (--especially if boys were still included here too). And you said 23% of 10 and 11 school year students didn’t answer (about 383 students if my calculation is correct), which is a pretty large percentage.

See response above.

• What was the ethnic breakdown of this survey? You control for it, but I’d be interested in seeing what the distribution looks like. Also, what does BME stand for?

See Table 1 in the new manuscript. BME is a term widely used in the UK that refers to ‘Black and Minority Ethnic’ group (essentially, the non-White part of the population). It is not necessarily our preferred terminology, though, so we have changed to White/non-White in the new manuscript.

Suggested grammatic and structural edits:

• MOPAC first appears in line 94 (page 5) and should be written out long form at that point, then abbreviated after that.

Amended

• The two sentences spanning lines 101-103 (page 5) “It is important to note… our analysis is therefore speculative.”) can probably be removed here and just be included in the limitations section of the Discussion.

Here we disagree with the reviewer – we think it’s important to signal upfront in the paper that our primary concern is theory-building.

• The sentence on lines 303-305 sounds a bit awkward (page 12) (“…Lauger (2014) found that…”)

Amended

• The sentence spanning line 333-336 (p. 13) (“Capturing gendered normative attitudes…) seems a bit out of place; it can probably wait until the Methods section when you talk about the data and measures used in greater detail.

Again we respectfully disagree with the reviewer, in as much as mentioning the survey items helps set up the second half of the paragraph, which covers the distinction between positive and negative social (and gendered) norms.

---

## [Decision Letter · Decision Letter 1]

21 Nov 2022

PONE-D-22-18323R1A street corner education: Stop and search, trust, and gendered norms among adolescent malesPLOS ONE

Dear Dr. Bradford,

Thank you for submitting your manuscript to PLOS ONE. After careful consideration, we feel that it has merit but does not fully meet PLOS ONE’s publication criteria as it currently stands. Therefore, we invite you to submit a revised version of the manuscript that addresses the points raised during the review process.

We look forward to receiving your revised manuscript.

Kind regards,

Guangyu Tong

Academic Editor

PLOS ONE

Journal Requirements:

Reviewers' comments:

Reviewer's Responses to Questions

**Comments to the Author**

1. If the authors have adequately addressed your comments raised in a previous round of review and you feel that this manuscript is now acceptable for publication, you may indicate that here to bypass the “Comments to the Author” section, enter your conflict of interest statement in the “Confidential to Editor” section, and submit your "Accept" recommendation.

Reviewer #1: (No Response)

Reviewer #2: All comments have been addressed

2. Is the manuscript technically sound, and do the data support the conclusions?

Reviewer #1: Yes

Reviewer #2: Yes

3. Has the statistical analysis been performed appropriately and rigorously? 

Reviewer #1: Yes

Reviewer #2: Yes

4. Have the authors made all data underlying the findings in their manuscript fully available?

Reviewer #1: Yes

Reviewer #2: Yes

5. Is the manuscript presented in an intelligible fashion and written in standard English?

Reviewer #1: Yes

Reviewer #2: Yes

6. Review Comments to the Author

Reviewer #1: The authors have largely addressed my concerns but I have a couple of small suggestions for what is, at this stage, a strong paper:

Line 103: I have to agree with my co-reviewer in not liking the language of “our analysis is therefore speculative.” Moreover, I think it sells the paper and it’s contributions short. Maybe refer to the paper as opening, rather than concluding a conversation? Refer to the findings as “generative”? This might be a case of subtle language differences between the US and the UK, but my read of “speculative” in this case is “not well grounded”, and I don’t think that’s the case with this paper.

Line 491: I appreciate the inclusion of table 1. However, can you split the true “no” answers on police contact from the “don’t know” responses, even if you need to combine them for your models? I find this breakdown itself to provide some insight into the kids’ legalization (I.e., how well they understand crime and police contact and how willing they are to discuss it.)

Reviewer #2: I want to thank the authors for their considerations of the reviews as well as the revisions they made. I appreciate the work the authors put into addressing the reviews provided to them and I am satisfied with their responses to my earlier feedback. I believe that this study makes an important and original contribution to the field.

7. PLOS authors have the option to publish the peer review history of their article (what does this mean?). If published, this will include your full peer review and any attached files.

Reviewer #1: No

Reviewer #2: No

---

## [Author Response · Author response to Decision Letter 1]

24 Nov 2022

We thank the reviewers for their very positive responses to the revised version of our article, and we are glad they feel it is in the position to make “an important and original contribution to the field” (R2). While R2 had no further comments to make, R1 did raise two final points, which we address below.

Line 103: I have to agree with my co-reviewer in not liking the language of “our analysis is therefore speculative.” Moreover, I think it sells the paper and it’s contributions short. Maybe refer to the paper as opening, rather than concluding a conversation? Refer to the findings as “generative”? This might be a case of subtle language differences between the US and the UK, but my read of “speculative” in this case is “not well grounded”, and I don’t think that’s the case with this paper.\\

We thank the reviewer for this comment, which is actually very positive, and we have amended the text at Line 103 along the lines they suggest: “our analysis should therefore be considered generative in nature”.

Line 491: I appreciate the inclusion of table 1. However, can you split the true “no” answers on police contact from the “don’t know” responses, even if you need to combine them for your models? I find this breakdown itself to provide some insight into the kids’ legalization (I.e., how well they understand crime and police contact and how willing they are to discuss it.)

We have added the ‘don’t know’ (n=19) and ‘did not answer’ (n=2) responses to the stop and search question, and ‘don’t know’ (n=50) and ‘don’t want to say’ (n=36) responses to Table 1 on p.19. Note this means footnotes to the table are no longer required.

---

## [Decision Letter · Decision Letter 2]

8 Dec 2022

A street corner education: Stop and search, trust, and gendered norms among adolescent males

PONE-D-22-18323R2

Dear Dr. Bradford,

We’re pleased to inform you that your manuscript has been judged scientifically suitable for publication and will be formally accepted for publication once it meets all outstanding technical requirements.

Kind regards,

Angelo Moretti, Ph.D.

Academic Editor

PLOS ONE

Additional Editor Comments (optional):

Reviewers' comments:

Reviewer's Responses to Questions

**Comments to the Author**

1. If the authors have adequately addressed your comments raised in a previous round of review and you feel that this manuscript is now acceptable for publication, you may indicate that here to bypass the “Comments to the Author” section, enter your conflict of interest statement in the “Confidential to Editor” section, and submit your "Accept" recommendation.

Reviewer #1: All comments have been addressed

Reviewer #2: All comments have been addressed

2. Is the manuscript technically sound, and do the data support the conclusions?

Reviewer #1: (No Response)

Reviewer #2: Yes

3. Has the statistical analysis been performed appropriately and rigorously? 

Reviewer #1: (No Response)

Reviewer #2: Yes

4. Have the authors made all data underlying the findings in their manuscript fully available?

Reviewer #1: (No Response)

Reviewer #2: Yes

5. Is the manuscript presented in an intelligible fashion and written in standard English?

Reviewer #1: (No Response)

Reviewer #2: Yes

6. Review Comments to the Author

Reviewer #1: (No Response)

Reviewer #2: I thank the authors for their additional revisions and appreciate the work they have done in this manuscript.

7. PLOS authors have the option to publish the peer review history of their article (what does this mean?). If published, this will include your full peer review and any attached files.

Reviewer #1: No

Reviewer #2: No

---

## [Editor Report · Acceptance letter]

15 Dec 2022

PONE-D-22-18323R2 

A street corner education: Stop and search, trust, and gendered norms among adolescent males 

Dear Dr. Bradford:

I'm pleased to inform you that your manuscript has been deemed suitable for publication in PLOS ONE. Congratulations! Your manuscript is now with our production department. 

Kind regards, 

on behalf of

Dr. Angelo Moretti 

Academic Editor

PLOS ONE